# Evaluate the stroke awareness of Palestinian undergraduate health students: A cross-sectional study on risk factors and warning signs

Mohammad Abuawad[1]*, Motaz Saifi[2,3], Maha Rabayaa[1,4], Mustafa Ghanim[1], Malik Alqub[1], Tartil Zarura[5], Diana Aldeek[5], Jomana Khateeb[5], Salah Abu Khalaf[5], Majdi Dwikat[6], Samar Alkhaldi[6], Johnny Amer[5], Ramzi Shawahna[1,7]

1 Department of Biomedical Sciences, Faculty of Medicine and Health Sciences, An-Najah National University, Nablus, Palestine, 2 Department of Internship, Ministry of Health, Ramallah, Palestine, 3 Department of Medicine, Faculty of Medicine and Health Sciences, An-Najah National University, Nablus, Palestine, 4 Department of Physiology, Faculty of Medicine, Bolu Abant İzzet Baysal University, Bolu, Turkey, 5 Faculty of Medicine and Health Sciences, An-Najah National University, Nablus, Palestine, 6 Department of Applied and Allied Medical Sciences, Faculty of Medicine and Health Sciences, An-Najah National University, Nablus, Palestine, 7 Clinical Research Center, An-Najah National University Hospital, Nablus, Palestine

* m.abuawad@najah.edu

## Abstract

### Background

Adequate knowledge of the clinical conditions associated with stroke, including risk factors and warning signs, is critical for improving healthcare outcomes. This study assesses undergraduate health students' knowledge and attitudes toward stroke, including risk factors and warning signs.

### Methods

A descriptive cross-sectional study was conducted with 1006 Palestinian undergraduate health students from several universities enrolled in medicine, pharmacy, nursing, and other health-related disciplines between March 2024 and July 2024. A validated self-administered questionnaire from previous publications was used to assess knowledge of stroke risk factors and warning signs. Attitudes were assessed using a Likert scale. The data were analyzed via SPSS 21 for Windows. Frequencies and percentages were used to summarize demographic data, whereas chi-square tests were performed to assess the associations between demographic characteristics and participants' knowledge scores (good/poor).

### Results

Among the 1006 participants, 55.4% were females, and 37.5% were in their third academic year. More than 65% of the participants were aware of the stroke-affected

**Data availability statement:** All relevant data are within the manuscript and its Supporting information files.

**Funding:** The author(s) received no specific funding for this work.

**Competing interests:** The authors have declared that no competing interests exist.

organ and the most affected gender. Awareness of key stroke risk factors including hypertension, smoking, alcohol consumption, diabetes mellitus, family history, previous strokes, obesity, and high cholesterol was high. The students' knowledge scores varied significantly by gender, academic year of study, place of residence, and field of education (p-value <0.05). Having good knowledge about stroke was significantly greater in females than in males (OR: 2.699), and was greater in sixth- and fourth-year students than in second-year students (OR: 6.855 and 1.704, respectively). However, it was significantly lower among health science students compared to preclinical medicine students (OR: 0.257).

## Conclusion

The current findings highlight the need for focused educational interventions, as medical students have greater awareness of stroke than their peers. Enhancing stroke awareness among health students is essential for facilitating early diagnosis and proper healthcare aid. Thus, targeting gaps in knowledge among health students should be part of broader efforts to improve stroke awareness.

## 1. Introduction

A stroke is a cerebral impairment resulting from either a vascular blockage (ischemic stroke) or bleeding (hemorrhagic stroke) that lasts more than 24 hours and potentially leading to death [1]. Studies have revealed that ischemic stroke is more common and accounting for approximately 87% of all stroke cases globally in 2016 [2,3]. Stroke incidence increases significantly with age. One in four individuals over the age of 25 is expected to experience a stroke in their lifetime [4]. Strokes are considered one of the leading causes of serious long-term disability [2]. Additionally, more than half of those who experience a stroke die as a result [4]. Stroke rates are increasing rapidly in low- and middle-income countries, where it is frequently more difficult for healthcare providers to deliver the care necessary for successful stroke prevention, treatment, and rehabilitation [4].

Understanding and addressing stroke risk factors can support early diagnosis, prevention, and reduction of stroke-related complications through timely hospitalization. Additionally, recognizing early warning signs is as important as knowing the risk factors for stroke. The early recognition of stroke has a crucial role in proper treatment and in improving clinical outcomes. As a result, educational programs about stroke warning signs that promote early detection of stroke are essential elements that can be disseminated through various media channels, including television, videos, newspapers, educational pamphlets, and health seminars [5]. Public awareness of stroke symptoms may speed up hospital arrival [6]. These symptoms generally include weakness, numbness, speech difficulties, dizziness, and vision loss. Each of these symptoms should be equally emphasized. Enhancing symptom recognition can significantly improve stroke survival rates and quality of life [7]. In recent years, interest in the potential value of of community-based stroke care services has increased [8].

Optimal patient care necessitates adherence to best practices that ensure adequate knowledge of clinical knowledge related to diseases, including evidence-based learning and continuous education, an integrated medical curriculum, active clinical training and simulation, an interdisciplinary approach, and patient-centered learning. Increased awareness of stroke can significantly influence morbidity and mortality rates while promoting healthier lifestyle choices [9]. To increase medical students' knowledge, attitudes, and practices regarding stroke, a more extensive educational program is essential. While previous studies have explored students' understanding of and awareness of stroke, a notable gap remains in the literature concerning the clinical presentations and awareness of stroke among undergraduate students [9]. The vital role of health sciences practitioners in the proper handling of stroke cases cannot be overlooked. Thus, targeting health students is highly valuable as they represent future healthcare professionals, and it is crucial to evaluate their knowledge of stroke. Therefore, this study aims to assess the knowledge of stroke risk factors and warning signs, and attitudes toward stroke patients among undergraduate healthcare students in Palestine, including medical students enrolled in human medicine programs and students in other undergraduate health-related disciplines.

## 2. Methodology

### 2.1. Study design and settings

This study employed a cross-sectional survey design to gather data from undergraduate health students across five universities in the West Bank of Palestine: (An-Najah National University, Al-Quds University, Palestine Polytechnic University, Arab American University, and Hebron University). A questionnaire survey was administered to students pursuing health-related studies for recruitment between 23 March 2024 and 11 July 2024. The objective of this study was to assess the knowledge of stroke risk factors and warning signs among undergraduate health students. This knowledge could guide the development of specialized educational interventions to enhance undergraduate health students' knowledge of stroke risk factors and warning signs, thereby improving their ability to identify and respond to stroke incidents proficiently.

### 2.2. Study population, inclusion criteria, and exclusion criteria

This research surveyed undergraduate health-related students from universities in the West Bank of Palestine. The students were in their second, third, fourth, fifth, and sixth years of study. Participants were divided into two categories: medical students, further split into preclinical and clinical phases, and students from other health-related programs. In addition to medical students, those in nursing, pharmacy, dentistry, and allied health sciences, were encouraged to participate. The general length of undergraduate programs in Palestine ranges from a minimum of four years for nursing and allied health sciences programs such as medical imaging, medical analysis, and physiotherapy; five years for pharmacy and dentistry programs; and six years for the medicine program. The exclusion criteria included students who were in their initial year of study in any of the aforementioned programs because they still had not taken any courses or discussed the topic previously in their curriculum, so they may have affected the results. Before participation, students were asked to provide informed consent. Those who refused to provide the informed consent, or did not complete all items in the questionnaire were excluded.

### 2.3. Ethical approval and consent to participate

The study protocol was authorized by the An-Najah National University Institutional Review Board (IRB), Nablus, Palestine (Ref: Med. March. 2024/2). Participants provided informed consent before they filled the questionnair. The informed consent form explained the objectives of the study, and assured the anonymity of the participants. The study was conducted in compliance with relevant guidelines and regulations.

### 2.4. Sample size and sampling techniques

Data collection occurred between 23–03–2024 and 11–07–2024, using a convenience sampling method. Using an online sample size calculation [10] with a reference proportion of 50%, a 95% confidence, and a 5% margin of error, the minimal

sample size was calculated to be 385. Ultimately, 1006 responses were collected and deemed eligible for analysis, as described in the flowchart (Fig 1). A large sample size was selected to improve the study's analytical power. Despite the calculated sample size was smaller than the chosen size, this decision was done to enhance the quality, reliability, and comprehension of the results. An electronic questionnaire was used to gather data from the students. The questionnaire was distributed to participants via electronic means, including university email systems, Facebook groups, WhatsApp groups, medical student forums, and researchers' social media accounts.

### 2.5. Data collection tool

The study tool was adapted from previously validated studies on the knowledge and attitudes of undergraduate health students about stroke [9,11]. The study questions were evaluated by specialists in the medical field and neuroscience academics to ensure their relevance to undergraduate students as well as their comprehensiveness and clarity. The

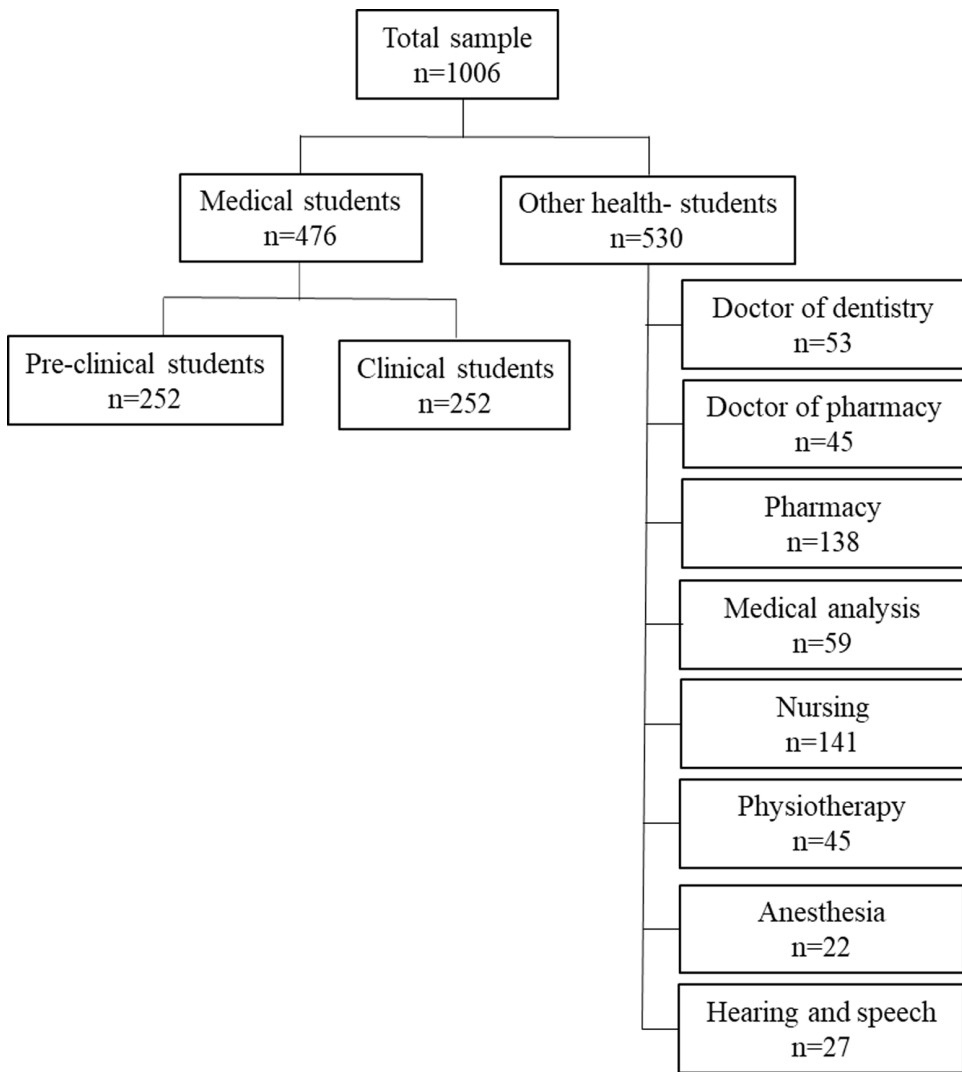

**Fig 1. Description of the sample based on the speciality.**

questionnaire consisted of three sections. The first section included the participants' demographic data: gender, marital status, place of residence, academic year, field of education, and cumulative academic score, which reflects the student's performance across multiple semesters or years, taking into consideration all the courses completed until the day of the survey, whether parents were working in the medical field, and their source of information about stroke. The second section included 19 questions to assess the students' knowledge about stroke. The questions included four general questions about stroke, 10 on stroke risk factors, and five on stroke warning signs. Each correct answer earned one point, with no points for incorrect answers. A score of 50% or higher was considered good, and while a score of <50% was considered poor. In the last section, the students were asked about their attitudes and recommendations to increase community awareness of stroke. The last section was descriptively analyzed, with the responses summarized as frequencies and percentages. Attitudes were assessed using a 3-point Likert scale (no, yes, or uncertain) to evaluate participants' views on stroke patients to lead a happy life and the role of families in patient recovery. The tool was tested for validity and reliability. Facial validity was ensured by reviewing the questionnaire by stroke specialists and health education experts. Content validity was ensured by modifying the tool and conducting a pilot study to ensure understanding and suitability. The pilot study involved 30 participants. The reliability of the study was evaluated, and found to be satisfactory (Cronbach's alpha = 0.72).

### 2.6. Statistical analysis

Statistical analyses were performed via the Statistical Package for the Social Sciences Statistics (SPSS) for Windows, version 21 (IBM Corp., Armonk, N.Y., USA). Descriptive statistics (frequencies and percentages) were applied for demographic characteristics, participants' attitudes toward stroke, and their suggestions for community awareness. Chi-square tests were used to determine the differences in general knowledge about stroke, risk factors, and warning signs among preclinical medicine students, clinical medicine students, and students from other health sciences. Additionally, the chi-square test was used to determine the associations between the knowledge scores (good/poor) of the participants and the demographic factors. Binary logistic regression was used to determine the degree of association between the significant variables from the chi-square test and good knowledge scores about stroke. A p-value of < 0.05 was considered statistically significant.

## 3. Results

### 3.1. Participants' demographic characteristics

The study included 1006 students studying medicine or health-related programs at one of the Palestinian universities. The majority of participants (55.4%) were females. The participants were in their second (24.8%), third (37.5%), fourth (21%), fifth (9.4%), or sixth (7.4%) academic year of study. Most of the participants had good or very good cumulative academic scores. The majority of the participants were single, and the majority were from either rural or urban areas. Twenty-five percent of the participants were in the preclinical phase-medicine program, 22.3% were studying the clinical phase-medicine program, and the remaining participants were studying other health-related programs. One fifth of the participants' parents were working in the medical field. The main source of information about stroke reported by the study participants was medical courses. The sociodemographic data of the students are presented in Table 1.

### 3.2. The variation in knowledge about stroke, risk factors, and warning signs across fields of study

As shown in Table 2, 74.1% of the participants knew that the brain is the affected organ in stroke, and 65.9% knew that stroke affects males more than females. However, fewer than 50% of the participants knew the meaning of stroke and the most affected age by stroke. More than 60% of the participants knew that hypertension, smoking, excessive alcohol consumption, diabetes mellitus, positive family history of stroke, previous stroke or TIA, and high cholesterol levels were

**Table 1. Demographic characteristics of the participants (n = 1006).**

| Variables | Frequency n (%) |
|---|---|
| **Gender** | |
| Male | 449 (44.6) |
| Female | 557 (55.4) |
| **Academic year** | |
| Second | 249 (24.8) |
| Third | 377 (37.5) |
| Fourth | 211 (21) |
| Fifth | 95 (9.4) |
| Sixth | 74 (7.4) |
| **Cumulative academic score** | |
| 0-- ≤ 1.7 Poor | 2 (0.2) |
| 1.7-- ≤ 2.35 Acceptable | 64 (6.4) |
| 2.35-- ≤ 3.00 Good | 462 (45.9) |
| 3.00-- ≤ 3.65 Very Good | 342 (34) |
| 3.65--4.00 Excellent | 136 (13.5) |
| **Marital status** | |
| Single | 975 (96.9) |
| Married | 25 (2.5) |
| Others | 6 (0.6) |
| **Place of residence** | |
| Urban | 500 (49.7) |
| Rural | 414 (41.2) |
| Camp | 92 (9.1) |
| **Field of education** | |
| Preclinical phase-medicine | 252 (25) |
| Clinical phase-medicine | 224 (22.3) |
| Doctor of Dental program | 53 (5.3) |
| Doctor of Pharmacy program | 45 (4.5) |
| Pharmacy program | 138 (13.7) |
| Medical analysis program | 59 (5.9) |
| Nursing program | 141 (14) |
| Physiotherapy program | 45 (4.5) |
| Anesthesia program | 22 (2.2) |
| Hearing and speech program | 27 (2.7) |
| **Do any of your parents work in the health field?** | |
| yes | 208 (20.7) |
| No | 798 (79.3) |
| **Source of information about stroke** | |
| 1. Social media and television | 278 (27.6) |
| 2. Medical courses | 742 (73.8) |
| 3. Academic research | 265 (26.3) |
| 4. Relative or personal experience | 181 (18) |
| 5. Textbook | 248 (24.7) |

considered risk factors for stroke. However, approximately one-third of the participants knew that migraines with aura and geographical ancestry were risk factors for stroke.

Additionally, Table 2 reveals significant differences in the students' knowledge of stroke based on the field of study, which was classified into three groups: preclinical medical students, clinical medical students, and other health sciences. The other health sciences group included the students studying health-related programs including the doctor of dental, doctor of pharmacy, pharmacy, medical analysis, nursing, physiotherapy, anesthesia, and hearing and speech programs as described previously in Table 1. Results revealed significant variations in all the general knowledge questions and warning signs among students who are studying preclinical medicine, clinical medicine, and other health sciences (p-value <0.05). Concerning the variation in knowledge about stroke risk factors in the field of study, significant variations were observed for all risk factors (p-value <0.05) except for geographical ancestry.

Compared with medical students, students studying other health sciences reported lower percentages of true answers for all of the general knowledge questions and warning signs. However, students studying other health sciences reported lower percentages of true answers for most but not all risk factors for stroke. In the clinical phase, medical students provided the lowest percentage of true answers regarding two of the risk factors: excessive alcohol consumption and having migraines with aura. The results are shown in Table 2.

**Table 2. The variation in knowledge of each of the general knowledge items about stroke, risk factors, and warning signs based on the participants' field of study (n = 1006).**

| | Total frequency n = 1006 (%) | Preclinical, medicine n = 252 (%) | Clinical, medicine n = 224 (%) | Other health sciences n = 530 (%) | p-value |
|---|---|---|---|---|---|
| **General knowledge about stroke (true answer)** | | | | | |
| 1. The affected organ (brain) | 745 (74.1) | 198 (78.6) | 193 (86.2) | 354 (66.8) | **<0.001** |
| 2. The meaning of stroke (thrombotic or hemorrhagic) | 462 (45.9) | 118 (46.8) | 152 (67.9) | 192 (36.2) | **<0.001** |
| 3. the most affected gender (male) | 663 (65.9) | 186 (73.8) | 149 (66.5) | 328 (61.9) | **0.004** |
| 4. the most affected age (>65 years) | 479 (47.6) | 148 (58.7) | 126 (56.3) | 205 (38.7) | **<0.001** |
| **Awareness about risk factors (true answer)** | | | | | |
| 1. Hypertension (yes) | 874 (86.9) | 235 (93.3) | 213 (95.1) | 426 (80.4) | **<0.001** |
| 2. Smoking tobacco products (yes) | 893 (88.8) | 229 (90.9) | 214 (95.5) | 450 (84.9) | **<0.001** |
| 3. Excessive alcohol consumption (yes) | 647 (64.3) | 179 (71) | 128 (57.1) | 340 (64.2) | **0.007** |
| 4. Diabetes (yes) | 703 (69.9) | 191 (75.8) | 194 (86.6) | 318 (60) | **<0.001** |
| 5. Family history of stroke (yes) | 745 (74.1) | 211 (83.7) | 193 (86.2) | 341 (64.3) | **<0.001** |
| 6. Having migraines with aura (yes) | 331 (32.9) | 96 (38.1) | 57 (25.4) | 178 (33.6) | **0.012** |
| 7. Previous stroke or transient ischemic attack (TIA) (yes) | 771 (76.6) | 225 (89.3) | 203 (90.6) | 343 (64.7) | **<0.001** |
| 8. Being overweight or obese (yes) | 793 (78.8) | 205 (81.3) | 197 (87.9) | 391 (73.8) | **<0.001** |
| 9. Having high cholesterol levels (yes) | 857 (85.2) | 224 (88.9) | 214 (95.5) | 419 (79.1) | **<0.001** |
| 10. Geographical ancestry (e.g., being African or Caribbean) (yes) | 333 (33.1) | 85 (33.7) | 68 (30.4) | 180 (34) | 0.611 |
| **Awareness about warning signs (true answer)** | | | | | |
| 1. Sudden numbness or weakness in the face, arm, or leg, especially on one side of the body (yes) | 817 (81.2) | 221 (87.7) | 190 (84.8) | 406 (76.6) | **<0.001** |
| 2. Sudden blurred or decreased vision in one or both eyes (yes) | 798 (79.3) | 214 (84.9) | 192 (85.7) | 392 (74) | **<0.001** |
| 3. Difficulty speaking or understanding speech (yes) | 843 (83.8) | 232 (92.1) | 199 (88.8) | 412 (77.70) | **<0.001** |
| 4. Severe headache with no known cause (yes) | 691 (68.7) | 186 (73.8) | 165 (73.7) | 340 (64.2) | **0.005** |
| 5. Dizziness or difficulty walking (yes) | 802 (79.7) | 219 (86.9) | 184 (82.1) | 399 (75.3) | **<0.001** |

### 3.3. Association between the knowledge score (poor/good) and the participants' demographic factors

Among the total study sample, 83% of the study participants achieved good knowledge about stroke. Table 3 shows the associations between the participants' demographic factors and having poor or good knowledge of stroke. The students' knowledge scores were significantly variable based on gender, academic year of study, place of residence, and field of education (p-value <0.05). However, the cumulative point score and marital status were nonsignificant factors.

### 3.4. Association between good knowledge scores and predictors

As demonstrated in Table 4, binary logistic regression was used to determine the predictor variables for achieving good knowledge about stroke. The score for achieving good knowledge was significantly higher in females than in males (OR=2.699, 95% CI = 1.874–3.888, p value<0.001). Compared with second-year students, sixth-year students achieved significantly greater knowledge about stroke (OR=6.855, 95% CI = 1.534–30.64, p-value = 0.012), followed by fourth-year students (OR=1.704, 95% CI = 1.039–2.795, p-value = 0.035). The score for achieving good knowledge was significantly lower among the students studying health sciences programs other than medicine (OR=0.257, 95% CI = 0.154–0.429; p-value<0.001).

**Table 3. Associations between the knowledge scores of the participants' demographic factors.**

| Variable | Knowledge score | | p-value |
|---|---|---|---|
| | Poor knowledge n = 171 (17%) | Good knowledge n = 835 (83%) | |
| **Gender** | | | |
| Male | 104 (23.2) | 345 (76.8) | **<0.001** |
| Female | 67 (12) | 490 (88) | |
| **Academic year** | | | |
| Second | 53 (21.3) | 196 (78.7) | **0.003** |
| Third | 61 (16.2) | 316 (83.8) | |
| Fourth | 42 (19.9) | 169 (80.1) | |
| Fifth | 13 (13.7) | 82 (86.3) | |
| Sixth | 2 (2.7) | 72 (97.3) | |
| **Cumulative point score** | | | |
| 0-- ≤ 1.7 Poor | 0 (0.0) | 2 (100) | 0.287 |
| 1.7-- ≤ 2.35 Acceptable | 8 (12.5) | 56 (87.5) | |
| 2.35-- ≤ 3.00 Good | 91 (19.7) | 371 (80.3) | |
| 3.00-- ≤ 3.65 Very Good | 51 (14.9) | 291 (85.1) | |
| 3.65--4.00 Excellent | 21 (15.4) | 115 (84.6) | |
| **Marital status** | | | |
| Single | 165 (16.9) | 810 (83.1) | 0.561 |
| Married | 4 (16) | 21 (84) | |
| Others | 2 (33.3) | 4 (66.7) | |
| **Place of residence** | | | |
| Urban | 78 (15.6) | 422 (84.4) | **0.047** |
| Rural | 69 (16.7) | 345 (83.3) | |
| Camp | 24 (26.1) | 68 (73.9) | |
| **Field of education** | | | |
| preclinical medicine | 23 (9.1) | 229 (90.9) | **<0.001** |
| clinical medicine | 18 (8) | 206 (92) | |
| other health sciences | 130 (24.5) | 400 (75.5) | |

**Table 4. Binary logistic regression for predictors of achieving good knowledge scores about stroke.**

| Variable (Reference) | B | S.E. | P-value | OR | 95% C.I. | |
| --- | --- | --- | --- | --- | --- | --- |
| | | | | | Lower | Upper |
| Female (male) | 0.993 | 0.186 | **<0.001** | 2.699 | 1.874 | 3.888 |
| Academic year (Second) | | | | | | |
| Third | 0.134 | 0.223 | 0.55 | 1.143 | 0.738 | 1.771 |
| Fourth | 0.533 | 0.252 | **0.035** | 1.704 | 1.039 | 2.795 |
| Fifth | 0.663 | 0.363 | 0.068 | 1.94 | 0.952 | 3.954 |
| Sixth | 1.925 | 0.764 | **0.012** | 6.855 | 1.534 | 30.64 |
| Place of residence (Urban) | | | | | | |
| Rural | 0.06 | 0.195 | 0.758 | 1.062 | 0.725 | 1.555 |
| Camp | -0.277 | 0.293 | 0.345 | 0.758 | 0.427 | 1.346 |
| Study program (preclinical medicine) | | | | | | |
| Clinical medicine | -0.204 | 0.362 | 0.573 | 0.816 | 0.401 | 1.657 |
| Other health sciences | -1.359 | 0.262 | **<0.001** | 0.257 | 0.154 | 0.429 |

## 3.5. Attitudes and recommendations for community awareness about stroke

As shown in Table 5, 25.3% of the study participants thought that stroke patients could not lead a happy life, and 74.3% of them thought that family care could help achieve earlier recovery of stroke patients after hospital discharge. Students were asked about suggestions for community education about stroke. A total of 61.7% of the participants suggested teaching the community about first aid, 50% suggested teaching the community about stroke symptoms and warning

**Table 5. Attitudes toward and recommendations for community awareness of stroke.**

| Attitude toward stroke | Frequency (%) |
| --- | --- |
| **Do you think people who have a stroke cannot lead a happy life?** | |
| No | 595 (59.1) |
| Yes | 255 (25.3) |
| Uncertain | 156 (15.5) |
| **Do you think family care for early recovery of stroke patients after hospital discharge?** | |
| No | 130 (12.9) |
| Yes | 747 (74.3) |
| Uncertain | 129 (12.8) |
| **Suggestions for community education** | |
| 1. First aid | 621 (61.7) |
| 2. Symptoms and warning signs | 503 (50) |
| 3. Risk factors | 481 (47.8) |
| 4. Prevention | 479 (47.6) |
| 5. I do not know | 75 (7.5) |
| **How to increase awareness about stroke among the community** | |
| 1. Involvement in stroke awareness campaigns | 411 (40.9) |
| 2. Giving awareness lectures | 589 (58.5) |
| 3. Writing posters | 356 (35.4) |
| 4. Through social media | 682 (67.8) |
| 5. No need for education | 34 (3.4) |

signs, 47.8% suggested teaching the community about stroke risk factors, and 47.6% suggested education about the prevention of stroke. Furthermore, the participants were asked about methods for increasing community awareness of stroke. Most of the participants chose awareness via social media (67.8%), followed by giving awareness lectures (58.5%), conducting stroke awareness campaigns (40.9%), and writing posters (35.4%).

## 4. Discussion

This study investigated the knowledge, attitudes, and recommendations about stroke among undergraduate healthcare students in Palestine. The study aimed to identify gaps in knowledge regarding stroke risk factors and warning signs to inform future educational interventions. The current findings revealed that 83% of students, particularly females, demonstrated a good level of knowledge about stroke. This finding is consistent with a previous study conducted among medical students in Saudi Arabia, which also reported a moderately high level of knowledge among medical students concerning stroke risk factors and symptoms of stroke [12]. Research has also demonstrated that most students can identify the brain as the affected organ and are aware of common risk factors of stroke such as hypertension and smoking. These findings align with previous reports [13–16]. Nevertheless, a study conducted among candidates for undergraduate medical entrance exams in Nepal revealed that only one-third of students recognized smoking as a risk factor of stroke [17]. Notably, there are significant disparities in knowledge levels across different fields of study, with medical students generally showing greater level of awareness than their counterparts in other health sciences. The findings of the current study are consistent with those of a previous investigation conducted among undergraduate healthcare students in Saudi Arabia [9]. Moreover, this study revealed that pharmacy students demonstrated a higher level of knowledge than nursing students.

The study revealed that medical students, particularly those in advanced years, demonstrated a higher level of stroke knowledge than their counterparts in other health disciplines, aligning with previous research conducted at Tabuk University in Saudi Arabia [14]. Nevertheless, the current study revealed students' knowledge gaps of the less common stroke risk factors, such as aura-related complications, in accordance with what was reported in a previous study at Tabuk university [14]. Moreover, this study provides new insights by revealing specific gaps in knowldege regarding less common risk factors such as excessive alcohol consumption, and the potential for stroke recovery. These findings are consistent with what was reported by a study in Saudi Arabia [18]. In contrast, a Turkish research revealed that over 77% of the students recognized alcohol as a stroke risk factor [19]. These findings suggest that, while the current curricula effectively educate healthcare students about common stroke indicators, there is a critical need to strengthen education about the less commonly addressed aspects of stroke. Bridging these gaps is essential to improve stroke management and prevention efforts across diverse healthcare settings.

Although clinical-phase medical students had the lowest proportion of correct responses regarding two risk factors, namely excessive alcohol consumption and migraines with aura, there were no significant differences between preclinical and clinical medical students in terms of achieving good knowledge about stroke based on logistic regression. The lower proportion of correct responses concerning alcohol consumption as a stroke risk factor may be explained by studies suggesting that light-to-moderate alcohol consumption is associated with a decreased risk of ischemic stroke [20,21]. Moreover, a previous study indicated that the relationship between alcohol consumption and stroke is influenced by genetic factors [20].

Notably, sixth-year students in the current sample achieved the highest scores in terms of knowledge scores. This is expected, as this group included only medical students in their final years, as other specialties in the study did not have six years in the curriculum. Sixth-year medical students are not only well trained in clinical practice, but also in the phase of preparation for exams, both for graduation and international exams for residency purposes. This finding is in line with previous studies that supported the idea of better basic medical knowledge by students in their clinical years as they use it in diagnosis [22,23]. The second-highest knowledge score was among fourth-year students. This could be attributed to the fact that these students are near their graduate year among other health specialities. For medical students, it is the first clinical year, and extensive exams for basic biomedical sciences are necessary to transition to the clinical phase.

Therefore, they still remember the information. The second-year medical students scored the best among the basic-year students, as courses of neuroanatomy, neurology and neurosciences are offered in the second year for medical and health sciences students in their curriculum. A similar finding was reported in a previous publication concerning students' better knowledge in genetics with the year in which they studied the genetics course [24].

Approximatly one quarter of the students (25.3%) in this study believed that stroke patients could not lead happy life. These findings could be explained by insufficient understanding of the role of rehabilitation in improving stroke patients' quality of life. Certain studies have shown the importance of professional reintegration through structured vocational programs in improving the quality of life and mental health status of stroke survivors [25,26]. Similarly, family involvement, which has been shown to be valuable support for post-stroke patients [27]. Further studies are needed to investigate this attitude among students and highlight the need for more comprehensive and targeted educational strategies to address these gaps ensuring that all healthcare students are adequately prepared to manage stroke in their future careers.

While this study provides valuable insights into the knowledge and awareness of stroke among Palestinian health students, it is important to acknowledge its limitations. The use of a convenience sampling method may limit the generalizability of the findings, as it might not fully represent the broader student population. Additionally, self-reported data might be subject to biases, such as social desirability or recall bias, which could influence the accuracy of the responses. Some questions were negatively framed and might have influenced participants' perceptions, causing a potential acquiescence bias. Despite these limitations, the results remain valid in addressing the research question, as the relative large sample size and the rigorous statistical analyses used help mitigate these potential biases. This reinforces the study's relevance and contribution to stroke awareness in health education. Therefore, while caution is needed in generalizing the results, the study still offers important insights that can inform future educational strategies and interventions.

To increase stroke education and awareness among health students, academic institutions should integrate comprehensive stroke content across all health-related programs, emphasizing both common and less-recognized risk factors, as well as rehabilitation potential, as recommended by a study in Taiwan [28]. Students' attitudes toward stroke can be effectively shifted through focused educational interventions, such as simulation-based training and structured engagement with stroke survivors [28]. Interdisciplinary training should be promoted to ensure that all healthcare providers - not just those in medicine - are well prepared to manage stroke. Structured curriculum enhancements, such as integrating stroke case scenarios in clinical training and encouraging collaboration between medical and allied health students, can ameliorate stroke education. Future research to explore the long-term impact of such educational interventions in Palestine should be investigated, like a previous study in India [29]. This could involve assessing the effectiveness of innovative teaching methods, such as simulation-based learning, alongside conducting cross-cultural comparisons to identify global best practices in stroke education. These steps will better equip future healthcare providers to address stroke prevention and management effectively. From a policy perspective, a revision to stroke education strategies should be considered by academic institutions in collaboration with healthcare institutions to ensure that students are sufficiently qualified in stroke management and prevention both theoretically and practically.

## 5. Conclusion

The study revealed that most participants, particularly females, medical students, and advanced-year students, demonstrated a good level of knowledge about stroke. However, significant gaps were found regarding less common risk factors of stroke, such as geographical ancestry and having a migraine with aura. These findings emphasize the need for focused educational interventions to improve knowledge about stroke risk factors and warning signs among undergraduate health students, particularly those studying health sciences programs other than medicine, as they have lower level of knowledge about stroke than medical students. To achieve thorough learning, educational programs should be adapted to different academic levels and disciplines due to knowledge gaps between students. Misconceptions about stroke patients' quality of life emphasize the need for empathy training and awareness in the curriculum.

## Supporting information

**S1 Data. Stroke awareness - Raw Data.**
(XLSX)

## Acknowledgments

The authors would like to thank An-Najah National University (www.najah.edu) for the technical support provided to publish the present manuscript. We would like to express our gratitude to Dr. Waleed Salameh, an expert in Educational English from the Faculty of Graduate Studies at An-Najah National University, for his invaluable assistance with the English editing of the revised manuscript.

## Author contributions

**Conceptualization:** Mohammad Abuawad, Motaz Saifi, Maha Rabayaa, Mustafa Ghanim, Malik Alqub, Majdi Dwikat, Samar Alkhaldi, Johnny Amer, Ramzi Shawahna.

**Data curation:** Tartil Zarura, Diana Aldeek, Jomana Khateeb, Salah Abu Khalaf.

**Formal analysis:** Mohammad Abuawad, Motaz Saifi, Maha Rabayaa, Mustafa Ghanim, Malik Alqub, Majdi Dwikat, Samar Alkhaldi, Johnny Amer, Ramzi Shawahna.

**Investigation:** Motaz Saifi, Maha Rabayaa.

**Methodology:** Mohammad Abuawad, Maha Rabayaa, Mustafa Ghanim, Malik Alqub, Majdi Dwikat, Samar Alkhaldi, Johnny Amer, Ramzi Shawahna.

**Supervision:** Mohammad Abuawad, Motaz Saifi, Mustafa Ghanim, Malik Alqub, Majdi Dwikat, Samar Alkhaldi, Johnny Amer, Ramzi Shawahna.

**Writing – original draft:** Mohammad Abuawad, Motaz Saifi, Maha Rabayaa, Mustafa Ghanim, Malik Alqub, Tartil Zarura, Diana Aldeek, Jomana Khateeb, Salah Abu Khalaf, Majdi Dwikat, Samar Alkhaldi, Johnny Amer, Ramzi Shawahna.

**Writing – review & editing:** Mohammad Abuawad, Motaz Saifi, Maha Rabayaa, Mustafa Ghanim, Malik Alqub, Majdi Dwikat, Samar Alkhaldi, Johnny Amer, Ramzi Shawahna.

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
