## [Decision Letter · Decision Letter 0]

17 Feb 2025

PONE-D-24-57604Evaluate the Stroke Awareness of Palestinian Undergraduate Health Students: A Cross-Sectional Study on Risk Factors and Warning SignsPLOS ONE

Dear Dr. Abuawad,

Thank you for submitting your manuscript to PLOS ONE. After careful consideration, we feel that it has merit but does not fully meet PLOS ONE’s publication criteria as it currently stands. Therefore, we invite you to submit a revised version of the manuscript that addresses the points raised during the review process.

We look forward to receiving your revised manuscript.

Kind regards,

Md. Shahjalal

Academic Editor

PLOS ONE

Journal Requirements:

2. Please include captions for your Supporting Information files at the end of your manuscript, and update any in-text citations to match accordingly. Please see our Supporting Information guidelines for more information: http://journals.plos.org/plosone/s/supporting-information .

Reviewers' comments:

Reviewer's Responses to Questions

**Comments to the Author**

1. Is the manuscript technically sound, and do the data support the conclusions?

Reviewer #1: No

Reviewer #2: Partly

2. Has the statistical analysis been performed appropriately and rigorously? 

Reviewer #1: Yes

Reviewer #2: Yes

3. Have the authors made all data underlying the findings in their manuscript fully available?

Reviewer #1: Yes

Reviewer #2: Yes

4. Is the manuscript presented in an intelligible fashion and written in standard English?

Reviewer #1: No

Reviewer #2: Yes

5. Review Comments to the Author

Reviewer #1: This is a good study regarding the assessment of knowledge and attitude of undergraduate students about stoke and its risk factors. However, there are some areas that require justifications and more detailed discussion to improve clarity in the manuscript.

Reviewer #2: 1. Unexpected higher awareness of stroke warning signs among preclinical students:

The findings suggest that preclinical medicine students had higher awareness on recognizing stroke warning signs as compared to students who had clinical training, which contradicts expectations. Since clinical medicine students have more experience and direct patient exposure, they should logically perform better in recognizing stroke symptoms. The authors should explicitly address this unexpected result in the discussion section.

2. Potential response bias due to negative question framing:

The question, "Do you think people who have stroke cannot lead a happy life?" is negatively framed, which may have influenced responses. Negative framing can lead to acquiescence bias, where participants are more likely to agree even if they are unsure. It could also cause priming where participants focus on challenges rather than recovery possibilities, potentially overestimating negative perceptions. The authors could consider discussing this potential response bias under the limitations section.

6. PLOS authors have the option to publish the peer review history of their article (what does this mean? ). If published, this will include your full peer review and any attached files.

**Do you want your identity to be public for this peer review?** For information about this choice, including consent withdrawal, please see our Privacy Policy .

Reviewer #1: No

Reviewer #2: **Yes: ** Aaryan Dahal

---

## [Author Response · Author response to Decision Letter 1]

24 Feb 2025

Response to reviewer is attached in submitting file.

---

## [Decision Letter · Decision Letter 1]

28 Mar 2025

PONE-D-24-57604R1Evaluate the Stroke Awareness of Palestinian Undergraduate Health Students: A Cross-Sectional Study on Risk Factors and Warning SignsPLOS ONE

Dear Dr. Abuawad,

Thank you for submitting your manuscript to PLOS ONE. After careful consideration, we feel that it has merit but does not fully meet PLOS ONE’s publication criteria as it currently stands. Therefore, we invite you to submit a revised version of the manuscript that addresses the points raised during the review process.

We look forward to receiving your revised manuscript.

Kind regards,

Md. Shahjalal

Academic Editor

PLOS ONE

Journal Requirements:

Reviewers' comments:

Reviewer's Responses to Questions

**Comments to the Author**

1. If the authors have adequately addressed your comments raised in a previous round of review and you feel that this manuscript is now acceptable for publication, you may indicate that here to bypass the “Comments to the Author” section, enter your conflict of interest statement in the “Confidential to Editor” section, and submit your "Accept" recommendation.

Reviewer #1: (No Response)

Reviewer #2: All comments have been addressed

2. Is the manuscript technically sound, and do the data support the conclusions?

Reviewer #1: No

Reviewer #2: Yes

3. Has the statistical analysis been performed appropriately and rigorously? 

Reviewer #1: Yes

Reviewer #2: Yes

4. Have the authors made all data underlying the findings in their manuscript fully available?

Reviewer #1: Yes

Reviewer #2: Yes

5. Is the manuscript presented in an intelligible fashion and written in standard English?

Reviewer #1: No

Reviewer #2: Yes

6. Review Comments to the Author

Reviewer #1: The paper is unclear regarding the needs of this paper in Palestinian setting and fails to deliver a coherent picture of the works done. The paper lacks coherence and the discussion is poor. Sampling and data collection is not robust.

Reviewer #2: This study has highlighted the gaps in the understanding of stroke risk factors among certain students. However, more explicit mention of how the medical and health science curriculum may be adjusted to improve stroke awareness among students would increase the value of this study. Similarly, how these findings could inform educational policy in Palestinian universities is needed. Overall, the manuscript is well-written, and addressing these points would strengthen its contribution to the field.

7. PLOS authors have the option to publish the peer review history of their article (what does this mean? ). If published, this will include your full peer review and any attached files.

**Do you want your identity to be public for this peer review?** For information about this choice, including consent withdrawal, please see our Privacy Policy .

Reviewer #1: No

Reviewer #2: **Yes: ** Aaryan Dahal

---

## [Author Response · Author response to Decision Letter 2]

2 Apr 2025

Dear editor and reviewer,

Reviewer 1: The paper is unclear regarding the needs of this paper in Palestinian setting and fails to deliver a coherent picture of the works done. The paper lacks coherence and the discussion is poor. Sampling and data collection is not robust.

Thank you for your recommendations. All the comments below have been elaborated as requested

Abstract:

1.The abstract starts with “A complete understanding” of..Im my view, understanding is a vague term and understanding something completely has no limits. Therefore, please use a suitable line to replace the statement, something that is more measurable.

Response: thanks for your comment. The sentence has been edited and become “ Adequate knowledge of clinical facts about stroke, including risk factors and warning signs is critical for improving health care outcomes”

1. This study aims to (you have written aimed to). Please consider writing in present tense as much as possible in the your abstract.

Response: thanks for your comment. The abstract has been elaborated as requested.

2. A survey “ questionnaire “, the word questionnaire is missing.

Response: thanks for your comment. The abstract has been elaborated as requested.

3. Please write how was attitute analysed or scored or what scale was used in your study.

Response: thanks for your comment. The following sentence has been added:” Attitudes were evaluated using a Likert scale”.

Additionally, the following statement is added to methods: ‘’The last section was descriptively analyzed, with responses summarized as frequencies and percentages. Attitudes were reported on a 3-point Likert scale (no, yes, or uncertain) to evaluate the attitude of participants regarding the ability of stroke patients to lead a happy life and the role of families in patient recovery.’’

4. The results in your abstract is too long. Please highlight major findings only in your abstract.

Response: thanks for your comment. The result section has been elaborated as requested.

Introduction:

1. Line 64 what are the best practices? Please mention.

Response: Thanks for your comment. The phrase has been edited and elaborated as requested. And become as follow “ Optimal patient care necessitates adherence to best practices that ensure adequate knowledge of clinical facts pertaining to diseases, including evidence-based learning and continous education, an integrated medical curriculum, active clinical training and simulation, an interdisciplinary approach, and patient-centered learning.”

2. Please include your definition of medical and other undergraduates as operationalized in your study to introduce your study population.

Response: Thank you for your comment. The study population is explained in the last paragraph of the introduction.

3. Line 102, how many universities ?

Thanks for your comment. The sentence has been edited and became as follow: “This study employed a cross-sectional survey design to gather data from undergraduate health students across five universities (An-Najah National University, Al-Quds University, Palestine Polytechnic University, Arab American University, and Hebron University) located in the West Bank of Palestine”

4. Line 103, who developed the questionnaire, team work? You hav only mentioned from previous publications, please provide some context of those papers as well.

Response: thank you for your comment. The study tool was adapted from previously validated studies about the knowledge and attitude of undergraduate health students about stroke (references 9, 11). The study questions were then evaluated by specialists in the medical field and academics in neuroscience to ensure the properness of the study questions for undergraduate students and the comprehensiveness and understandability of the questions.

5. Line 107 .It can contribute…, please mention how to add more clarity.

Response: thanks for your comment. The sentence has been rephrased as follow: “This knowledge could direct toward the development of specialized educational interventions aimed at strengthening undergraduate health students' comprehension of stroke risk factors and warning signs, hence improving their capacity to identify and respond to stroke incidents proficiently.”

6. Line 113..and beyond, please refrain from using such vague words. Please mention how many years in exact number.

Response: thanks for your comment. The sentence has been elaborated as follow:” This research included a survey of undergraduate health-related students from universities located in the West Bank of Palestine. The students were in their second, third, fourth, fifth, and sixth years of study.”

Methodology

7. Please consider adding a table showing medical related streams and non medical related streams in a flowchart to give a comprehensive picture of how many were selected from which stream.

Response: thanks for your comment. the following flowchart in methods has been made and added in the methods as figure 2.

8. Why was 1006 samples used and probability sampling technique not applied if we has such a large sampling frame?

Response: Thanks for your insightful question. The minimum sample size was 385, but we collected 1006 to improve statistical power and generalizability. The greater sample size reduces variability and better represents health student groupings. Convenience sampling was used to enhance the quality and reliability of the results and to provide a more thorough comprehension of the population. Probability sampling proved impractical for the study's time and resources. We included students from numerous colleges and academic years to ensure diversity. Although non-random, this method reveals students stroke knowledge and attitudes.

9. Please add the map after your study site section in the methodology part.

Response: Thanks for your comment. The figure has been inserted as requested under study design and setting section in the methodology.

10. Again, please elaborate how the “attitude” variable was interpreted or only knowledge was interpreted, was it only presented descriptively?

Response: thanks for your comment. The following phrase has been added “Descriptive statistics (frequencies and percentages) were applied for demographic characteristics, participants’ attitudes toward stroke.”

Discussion

Please mention discussions regarding the attitude variable as well. Please add lines regarding how and in what ways attitude can be changed regarding stroke in Palestinian undergraduates. The discussion is very simple and needs to be significantly improved with strong recommendations for policy makers.

Response: thanks for your comment. Edited, the following phrases has been added to the discussion part:

“Nearly one quarter of the students (25.3%) in this study believed that stroke patients could not lead a happy life. These findings could be explained by an insufficient understanding of the role of rehabilitation in improving stroke patients' quality of life. Certain studies have shown the importance of professional reintegration with structured vocational programs in improving the quality of life and mental health status of stroke survivors (26,27). Similarly for family involvement, which has been shown to be a valuable support for post-stroke patients (28).”

“Students’ attitudes toward stroke can be effectively shifted through focused educational interventions, such as simulation-based training and structured engagement with stroke survivors (30). Interdisciplinary training should be promoted to ensure that all healthcare providers, not just those in medicine, are well prepared to manage stroke. Additionally, structured curriculum adjustments, such as including stroke case scenarios in clinical training and encouraging collaboration between medical and allied health students, could help to further support this.”

“From a policy perspective, a revision to stroke education strategies should be considered by academic institutions in collaboration with healthcare institutions to ensure that students are exposed to stroke management and prevention theoretically and practically.”

Overall: The manuscript lacks clarity at many parts. Please consider showing your work to a person fluent in English language.

Response: done. thanks for the comment.

Reviewer 2: This study has highlighted the gaps in the understanding of stroke risk factors among certain students. However, more explicit mention of how the medical and health science curriculum may be adjusted to improve stroke awareness among students would increase the value of this study. Similarly, how these findings could inform educational policy in Palestinian universities is needed. Overall, the manuscript is well-written, and addressing these points would strengthen its contribution to the field.

Response: Edited, thanks for the comment. The following has been added in the discussion part.

“Students’ attitudes toward stroke can be effectively shifted through focused educational interventions, such as simulation-based training and structured engagement with stroke survivors (30). Interdisciplinary training should be promoted to ensure that all healthcare providers, not just those in medicine, are well prepared to manage stroke. Additionally, structured curriculum adjustments, such as including stroke case scenarios in clinical training and encouraging collaboration between medical and allied health students, could help to further support this.”

“From a policy perspective, a revision to stroke education strategies should be considered by academic institutions in collaboration with healthcare institutions to ensure that students are exposed to stroke management and prevention theoretically and practically.”

---

## [Decision Letter · Decision Letter 2]

15 Apr 2025

PONE-D-24-57604R2Evaluate the Stroke Awareness of Palestinian Undergraduate Health Students: A Cross-Sectional Study on Risk Factors and Warning SignsPLOS ONE

Dear Dr. Abuawad,

Thank you for submitting your manuscript to PLOS ONE. After careful consideration, we feel that it has merit but does not fully meet PLOS ONE’s publication criteria as it currently stands. Therefore, we invite you to submit a revised version of the manuscript that addresses the points raised during the review process.

We look forward to receiving your revised manuscript.

Kind regards,

Md. Shahjalal

Academic Editor

PLOS ONE

Journal Requirements:

Reviewers' comments:

Reviewer's Responses to Questions

**Comments to the Author**

1. If the authors have adequately addressed your comments raised in a previous round of review and you feel that this manuscript is now acceptable for publication, you may indicate that here to bypass the “Comments to the Author” section, enter your conflict of interest statement in the “Confidential to Editor” section, and submit your "Accept" recommendation.

Reviewer #1: All comments have been addressed

Reviewer #2: All comments have been addressed

2. Is the manuscript technically sound, and do the data support the conclusions?

Reviewer #1: Yes

Reviewer #2: Yes

3. Has the statistical analysis been performed appropriately and rigorously? 

Reviewer #1: Yes

Reviewer #2: Yes

4. Have the authors made all data underlying the findings in their manuscript fully available?

Reviewer #1: Yes

Reviewer #2: Yes

5. Is the manuscript presented in an intelligible fashion and written in standard English?

Reviewer #1: No

Reviewer #2: Yes

6. Review Comments to the Author

Reviewer #1: Please improve your English language. Technically, the paper has improved so much from its last submission but the re are some language and grammar related issues can be improved to enhance the readability and clarity of your paper.

Reviewer #2: The submission is well presented and acceptable for publication. All comments have been thoroughly addressed.

7. PLOS authors have the option to publish the peer review history of their article (what does this mean? ). If published, this will include your full peer review and any attached files.

**Do you want your identity to be public for this peer review?** For information about this choice, including consent withdrawal, please see our Privacy Policy .

Reviewer #1: No

Reviewer #2: **Yes: ** Aaryan Dahal

---

## [Author Response · Author response to Decision Letter 3]

17 Apr 2025

Dear editor and reviewer,

Thank you for your time and your valuable comments which we believe they helped us in improving the quality of our manuscript.

Kind regards,

Corresponding author

6. Review Comments to the Author

Reviewer #1: Please improve your English language. Technically, the paper has improved so much from its last submission but the re are some language and grammar related issues can be improved to enhance the readability and clarity of your paper.

Response: thanks for your comment. The English language has been edited all over the manuscript as highlighted in red in the revised manuscript by authors and with the help of an English expert from our university to whom we add acknowledgment at the end of the manuscript.

Reviewer #2: The submission is well presented and acceptable for publication. All comments have been thoroughly addressed.

Response: thanks for your recommendation

---

## [Editor Report · Decision Letter 3]

22 Apr 2025

Evaluate the Stroke Awareness of Palestinian Undergraduate Health Students: A Cross-Sectional Study on Risk Factors and Warning Signs

PONE-D-24-57604R3

Dear Dr. Mohammad,

We’re pleased to inform you that your manuscript has been judged scientifically suitable for publication and will be formally accepted for publication once it meets all outstanding technical requirements.

Kind regards,

Md. Shahjalal

Academic Editor

PLOS ONE

Additional Editor Comments (optional):

Thanks for addressing all comments appropriately. 
---

## [Editor Report · Acceptance letter]

PONE-D-24-57604R3

PLOS ONE

Dear Dr. Abuawad,

I'm pleased to inform you that your manuscript has been deemed suitable for publication in PLOS ONE. Congratulations! Your manuscript is now being handed over to our production team.

Kind regards,

on behalf of

Dr. Md. Shahjalal

Academic Editor

PLOS ONE